# Measles Immunization Status of Health Care Workers: A Cross-Sectional Study Exploring Factors Associated with Lack of Immunization According to the Health Belief Model

**DOI:** 10.3390/vaccines11030618

**Published:** 2023-03-09

**Authors:** Vincenzo Restivo, Alessandra Fallucca, Federica Trapani, Palmira Immordino, Giuseppe Calamusa, Alessandra Casuccio

**Affiliations:** Department of Health Promotion, Maternal and Infant Care, Internal Medicine and Medical Specialties (PROMISE) “G. D’Alessandro”, University of Palermo, Via del Vespro 133, 90127 Palermo, Italy

**Keywords:** healthcare workers, vaccination refusal, vaccine, hesitant, health belief model, measles, communication, barrier, awareness, immunization

## Abstract

Suboptimal levels of measles vaccination coverage make Italy a country where the circulation of the virus is still endemic. In the past decade, several nosocomial outbreaks of measles occurred in Italy that rapidly spread the infection among large numbers of hospitalized patients and susceptible healthcare workers (HCWs). A cross-sectional study was conducted at the University Hospital of Palermo (Italy) to estimate the rate of HCWs immunization and to investigate the factors associated with lack of immunization. The attitude to the immunization practice was evaluated by exploring the Health Belief Model. Overall, 118 HCWs were enrolled, with a mean age of 31 years and 59.3% male. About half of the sample (45.8%, *n* = 54) was found not to be immunized against measles. Multivariable analysis showed that the factors directly associated with the non-immunization status against measles were female sex (OR = 3.70, *p* = 0.056), being an HCW different from a physician (OR = 10.27, *p* = 0.015), having a high perception of barriers to vaccination (OR = 5.13, *p* = 0.047), not being immunized for other exanthematous diseases such as chickenpox (OR = 9.93, *p* = 0.003), mumps (OR = 33.64, *p* < 0.001) and rubella (OR = 10.12, *p*= 0.002). There is a need to contrast the low adherence of HCWs to measles vaccination by identifying effective strategies to increase immunization coverage and limiting the risk of further nosocomial measles outbreaks.

## 1. Introduction

Measles is a highly contagious vaccine-preventable disease that can cause serious complications in susceptible people. In the World Health Organization (WHO) European Region, measles endemic transmission was stopped in 37 out of 53 Member States (MS) by 2018. Sixteen MS (30%) are still measles-endemic, including Italy [1]. From the beginning of 2017 to the end of 2018, 8078 cases of measles were reported in Italy, of which 5397 in 2017 and 2681 in 2018, with an incidence of 89.1 and 44.3 cases per million inhabitants, respectively [2]. The emergence of measles outbreaks in Italy is a multifaceted phenomenon due to a variety of factors. However, there is growing evidence that the main cause is the accumulation of a large pool of susceptible populations due to low vaccination uptake over the years [3]. Vaccination coverage shows a decreasing trend, falling below the 95% threshold needed to stop the circulation of the virus, according to the recommendations of the WHO and the Italian National Plan for the Elimination of Measles and Rubella. In Italy, measles vaccination coverage in 2020 was 92.8% for the first dose and 88.3% for the second one among the 18 years old people [4]. The combined measles-mumps-rubella vaccine was introduced in Italy in the 1990s, and universal vaccination was recommended for all newborns [5]. The current Italian Immunization Program for measles provides a first dose of vaccine at 13–15 months of life and a second one at 6 years. Since 2017, vaccination against measles has been included in “mandatory childhood vaccinations,” but there is no vaccination catch-up policy for non-immunized adults [6].

The endemic circulation of measles can lead to several outbreaks of the disease also in the healthcare setting, as occurred in Italy [7,8]. Indeed, measles’ high effective reproduction number and virus spread kinetics increase the risk of large-scale outbreaks in hospitals [8,9]. Measles can spread rapidly in nosocomial settings causing a major public health emergency, as non-immune healthcare workers (HCWs) can contract the infection due to the high volume of contacts. It is estimated that the risk of acquiring pathologies such as measles is estimated to be two to 19 times higher in HCW than in the general population [10]. Moreover, HCWs could transmit the infection to patients that are at high risk of severe outcomes. Currently, although in Italy, vaccination for HCWs is strongly recommended by the National Vaccine Prevention Plan (PNPV) as a useful preventive practice, there are no regulatory restrictions on the employment of HCWs based on immune status monitoring [6,11].

Although the measles vaccine has been available in Italy for over 20 years, and its safety and efficacy have been demonstrated over time, vaccine hesitancy is also present among HCWs [12,13]. The WHO identified the “reluctance or refusal to vaccinate despite the availability of vaccines” as one of the 10 threats to global health in 2019 [13]. Notwithstanding, a growing number of people perceive vaccines as unsafe and unnecessary [14]. Vaccine decision-making can be explained by cognitive models such as the Health Belief Model (HBM) [15]. Developed in the 1950s, HBM continues to be one of the most accredited and widely used theories to investigate the attitude to adhere to a preventive practice, such as vaccination. HBM allows us to predict adopted behaviors by exploring, on the one hand, people’s perceived susceptibility or vulnerability to a certain disease and, on the other hand, the expected potential benefits and barriers to vaccination [16,17]. Vaccine decision-making may also be influenced by social, demographic and economic factors that could be the determinants of measles vaccination uptake [18,19].

The main objective of this study is to investigate the lack of immunization against measles among HCWs. The second objective is to investigate HCWs’ perceptions of the severity of this exanthematous disease and of the efficacy of vaccination as a strategy to prevent it in a country where nosocomial outbreaks of measles are documented.

## 2. Materials and Methods

A cross-sectional study was conducted at the University Hospital of Palermo, Italy, a large teaching hospital with more than 600 beds in various wards (predominantly medical and surgical specialties), from January to May 2022, through the administration of a questionnaire addressed to HCWs. The study population included all HCWs for which a telephone number and email address were available in the register of the University Hospital of Palermo. About 2000 HCWs work at the University Hospital of Palermo; we invited the HCWs with an available telephone number and email address (*n* = 900) to respond to the questionnaire. The sample was recruited by sending text messages and emails containing a description of the objectives and methods of the study, an informed consent form and a link to an anonymous questionnaire.

The questionnaire consisted of 4 sections. The first concerned socio-demographic information and personal behaviors with items about age, sex, smoking and alcohol habits, working activities and healthcare setting. The second section consisted of the HBM that investigates, through 8 items, four different domains of the model: benefits related to measles vaccinations, barriers for acceptability of measles vaccine, susceptibility of infection and severity of diseases. The available response options, according to a Likert scale, ranged from 1 = “I strongly disagree” to 5 = “I strongly agree.” For each respondent, the scores relating to the 2 items of the same domain were added (range score 2–10 per domain), and the median value was calculated (median score = 8). The median value was used as a cut-off to recategorize the score for each domain as “High level” ≥ 8 or “Medium-Low level” < 8. The third section of the questionnaire investigated the attitudes and beliefs of HCWs towards vaccinations as follows, considering: vaccinations for healthcare professionals as a prerequisite for their health, vaccinations of health workers to avoid absences from work, and recommending vaccinations to patients in clinical practice. The fourth one investigated the immunization status against measles and other exanthematous diseases, such as mumps, rubella and chickenpox, the natural acquisition of the infection and the eventual intention to recover the vaccination against measles not done previously. Vaccination status was ascertained by checking the Vaccination Register of the Palermo Hospital Vaccination Unit, which was based on vaccinations reported on vaccination cards.

The study was conducted in accordance with the Declaration of Helsinki and approved by the Ethics Committee Palermo 1 (protocol code 09/2021 on 15 September 2021).

The normality of the distribution for the quantitative variables was assessed with the Skewness and Kurtosis test. Means and standard deviations (SDs) were chosen for the normal distribution of these variables, while median and interquartile range (IQR) was used for non-normal distribution. The absolute and relative frequencies were calculated for the qualitative variables. The differences of quantitative variables normally and not normally distributed among measles immunization status were evaluated respectively with Student’s T-test and the Wilcoxon and Mann–Whitney tests, while for the qualitative variables, the Chi^2^ test was used. Univariable logistic regression analysis was performed to evaluate the factors associated with the lack of measles immunization. A multivariable logistic regression model was used to analyze the covariates associated with univariable analysis with a *p*-value lower than 0.10 and for a priori confounding variables. For all analyses, a *p*-value of 0.05 was assumed to indicate significance (2-tailed). All collected data were analyzed using Stata/SE 14.2 statistical software (Copyright 1985–2015, StataCorp LLC, 4905 Lakeway Drive, College Station, TX, USA. Revision 29 January 2018).

## 3. Results

Overall, 120 out of 900 HCWs contacted by telephone and email were enrolled (response rate = 13%), but 2 were excluded because they did not consent to the processing of personal data. The majority of people interviewed were male (59.3%, *n* = 70), and the mean age was 31 years old. A considerable number of respondents (45.8% *n* = 54) were not immunized against measles because they did not contract the natural infection and did not receive the two doses of vaccine needed for protection. With regards to employment, 33.1% (*n* = 39) were medical residents, followed by healthcare students (30.5%, *n* = 36), physicians (19.5%, *n* = 23), and other healthcare professionals (16.9%, *n* = 20). About a quarter of HCWs suffered from a chronic, acute, or allergic condition at the time of the interview (25.6%, *n* = 29) and about a third reported taking medications regularly (33.1%, *n* = 39). More than half of the sample reported alcohol consumption two to four times a month, and 26.3% (*n* = 31) said they smoked regularly (Table 1).

The HBM construct analysis showed that 72.9% (*n* = 86) of HCWs had a high level of risk perception related to measles infection, and 81.4% (*n* = 96) correctly perceived the benefits of measles vaccination, but, at the same time, more than a third (39.8%, *n* = 47) believed that there were difficulties and barriers related to vaccination adherence. Furthermore, investigating the immune status against other vaccine-preventable viral diseases, it was found that the majority of respondents were immunized against chickenpox (77.9%, *n* = 92), mumps (60.2%, *n* = 71) and rubella (57.6%, *n* = 68; Table 2).

Comparison between “non-immunized” and “immunized” HCWs for measles showed that the non-immunized were more frequently female (50.0% vs. 32.8%, *p* = 0.058) and medical students (37.0% vs. 25.0%, *p* = 0.301; Table 1). Additionally, non-immunized HCWs more frequently reported that it was not their competence to recommend vaccination to patients (24.1% vs. 6.3%, *p* = 0.070), had low perceptions of measles-related severity (42.6% vs. 28.1%, *p* = 0.100) and had a low perception of vaccination benefits (20.4% vs. 17.2%, *p* = 0.658; Table 2).

At the multivariable analysis, the factors associated with the non-immunization status against measles were: being an HCW different from a physician (OR = 10.27, *p* = 0.015), high perception of barriers to vaccination (OR = 5.13, *p* = 0.047), not being immunized for other exanthematous diseases such as chickenpox (OR = 9.93, *p* = 0.003), mumps (OR = 33.64, *p* < 0.001) and rubella (OR = 1.12, *p* = 0.002; Table 3).

## 4. Discussion

Nosocomial transmission of measles represents a major and emerging public health problem that seriously threatens immunization aimed at achieving the elimination of vaccine-preventable diseases. The measles outbreaks that have occurred in several Italian healthcare facilities were signals of the resurgence risk of measles due to the reduction in the level of immunization against measles, both among the general population and HCWs [6,7].

The present study highlighted a rather low level of protection against measles, revealing that about half of the HCWs interviewed had neither received two doses of the vaccine nor acquired natural immunity. The immunization rate of the recruited HCWs was similar to what has been reported by other studies carried out in the Italian healthcare context [20]. However, the level of protection against measles of Italian HCWs is lower than in other European countries [21]. The crucial role of HCWs in measles transmission has been demonstrated, given the potential for rapid and large-scale spread of the disease in healthcare settings [22]. The inadequate HCWs immunization rate could be among the main causes of the endemic measles circulation in Italy, which is, in fact, among the 16 WHO European Regions that have not yet succeeded in stopping measles [1,10].

In Italy, measles vaccination was introduced in 1979 in the pediatric population, and later, in the 1990s, the combined vaccine against measles-mumps-rubella was approved for all newborns and recommended for all adults of categories at risk of contagion and transmission by occupational exposure [23]. Furthermore, Italian law 119/2017, containing “Urgent provisions on vaccine prevention, infectious diseases and disputes relating to the administration of drugs,” and the Ministerial Circular 25233/2017 ordered the collection of data on the vaccination and immunological status of workers in some settings exposed to a high number of contacts, such as school, health, and social care [5,24]. In fact, as required by the decree of the President of the Italian Republic 445/2000, teachers, social workers, and HCWs must submit a report of their vaccination status to the institutions for which they work [25]. However, despite these legal measures, vaccinations have not become mandatory for HCWs, and therefore, they have no legal obligation to be vaccinated. One approach to addressing this issue could be to ensure that measles vaccination becomes a condition of pre-placement authorization for HCWs or, alternatively, to demonstrate infection by serological screening [26]. However, the mandatory nature of vaccination for workers in some specific risk sectors remains a controversial issue [27]. HCWs should not be inclined to accept vaccination due to the risk of infecting themselves during care practices but for the professional responsibility to the care of their patients. Despite this, a substantial percentage of respondents to this survey, nearly one-fifth, believed that measles vaccination for HCWs was not essential and should not be mandatory. These data are quite critical, considering a high percentage of HCWs are not immunized for measles and other vaccine-preventable diseases [28,29]. Awareness campaigns conducted on-site and with specific one-to-one counseling for hesitant HCWs, may be needed to counter measles vaccine refusal [30].

This study showed that physicians had less probability of not being immunized against measles than other professional roles in health settings. Similarly, previous surveys about immunization attitudes and risk perceptions of rash diseases in HCWs revealed that physicians were more immunized to conditions such as chickenpox and mumps and that they had higher vaccine knowledge and a higher likelihood to be vaccinated [31,32]. One possible explanation of this evidence should be that the medical formation acquired through the medicine course could lead to more complete and in-depth training on infectious diseases and their possible complications, making physicians more aware of the importance of vaccinations [33]. Training courses focused on prevention and vaccination should be offered to all HCWs in order to increase knowledge and awareness of vaccine-preventable diseases and to increase acceptance of measles vaccination.

A factor associated with not immunization against measles was the high perception of barriers to vaccination. This factor, belonging to the HBM constructs, was investigated with items regarding accessibility to the vaccination centers and their opening hours. Most not immunized HCWs and the general population perceived the opening hours of vaccination clinics as an obstacle to getting vaccinated [34,35]. This data could be due to the working hours of HCWs, which very often involve double shifts or fast shifts with little free time between the transition of two shifts. As demonstrated by a study conducted in Norway, long work shifts are very frequent in the health sector and are associated with considerable difficulties in adopting a healthy lifestyle, which would also include prevention through vaccination [36]. Increasing the accessibility of vaccinations, such as offering vaccinations directly in the workplace, could be a solution to implement vaccine acceptability among not immunized HCWs.

Another factor significantly associated with not immunization status against measles was the not immunization status against other exanthematous diseases, such as chickenpox, mumps, and rubella. The lack of immunization against these diseases was due to the lack of vaccination and the lack of immunity acquired following a natural infection. Several studies have shown that people who get a viral infection have a higher risk of getting secondary infections. The measles virus, in particular, has been identified as a suppressor of the immune system, and the authors attribute the immunosuppression to the depletion of T and B lymphocytes [37,38]. Immune suppression is, therefore, a possible cause of the relationship between multiple infectious diseases, but it cannot be excluded that other factors may also contribute. It is known that some behavioral habits play an important role in the prevention of infectious diseases, such as hand washing, the use of medical-surgical masks, and social distancing [39]. Therefore, the non-occurrence of measles infection and other infectious diseases caused by respiratory viruses could also be related to the adoption of these hygienic-sanitary standards [40]. However, the adoption of health and hygiene standards cannot be sufficient to protect HCWs and prevent them from infecting hospitalized and frail patients due to the high number of contacts in the healthcare setting. It is necessary to reiterate the importance of vaccination against measles and to contrast vaccine hesitancy among healthcare professionals, promoting clear and effective communication about vaccinations and adopting innovative strategies that lead to informed acceptance of vaccination, such as targeted training for HCWs.

A result with a marginal statistical significance that emerged from the analysis concerns the lower immunization rate for measles in women compared to men. This result is confirmed by a multicenter cross-sectional study conducted in Italy but contrasts with the data more frequently reported in the literature, where the higher rates in women are due to the prevention of risks related to some infections in pregnant women [20,41]. These results could represent a first alert on the change in women’s attitude towards the vaccination practice, which could reflect a decrease in women’s attention to prevention in pregnancy.

The main limitation of this study is the recall bias due to the delay between measles vaccination practice and questionnaire administration. Furthermore, the immunological status of healthcare workers was not verified with serological tests, but the immunization status against measles was self-reported by the respondents because the Sicily region does not have a single vaccination registration system. Furthermore, the enrolment procedure based on SMS and e-mail could have included HCWs more suitable to answer online questionnaires, and the small sample size is not representative of all Italian HCWs. This could limit the generalizability of data. Despite the previous limitations, this study explored the association between measles immunization and one of the well-established cognitive behavioral models, such as the Health Belief Model, in a population of HCWs with suboptimal measles immunization coverage.

## 5. Conclusions

This study revealed inadequate levels of measles protection among HWCs related to the high risk of nosocomial outbreaks of infection. Refusal of measles vaccination among HCWs may be influenced by perceived barriers to vaccination and low awareness of disease severity. It may be necessary to identify innovative strategies for vaccinations offered to HCWs and promote training courses in order to sensitize HCWs to preventive practices and achieve optimal vaccination coverage for measles. Further studies are needed with a large HCWs population in order to confirm these results.

## Figures and Tables

**Table 1 vaccines-11-00618-t001:** Characteristics of enrolled HCW by measles immunization status.

		Total HCW Respondents	Measles Immunized *n*(%)	Measles Not Immunized *n*(%)	*p* Value
		118	64 (54.2%)	54 (45.8%)	
Mean Age		31.0	31.9	30	0.57
Gender					
	Male	70 (59.3%)	43 (67.2%)	27 (50.0%)	0.058
	Female	48 (40.7%)	21 (32.8%)	27 (50.0%)	
Occupation					
	Physician	23 (19.5%)	17 (26.6%)	6 (11.1%)	0.163
	Medical resident	39 (33.1%)	20 (31.3%)	19 (35.2%)	
	Medical student	36 (30.5%)	16 (25.0%)	20 (37.0%)	
	Other HCW	20 (16.9%)	11 (17.2%)	9 (16.7%)	
Chronic or acute or allergic disease					
	Yes	29 (25.6%)	16 (25.0%)	13 (24.1%)	0.907
	No	89 (75.4%)	48 (75.0%)	41 (75.9%)	
Taking medications regularly					
	Yes	39 (33.1%)	22(34.4%)	17 (31.5%)	0.709
	No	79 (66.9%)	42 (65.6%)	37 (68.5%)	
Smoke					
	Yes	31 (26.3%)	15 (23.4%)	16 (29.6%)	0.446
	No	87 (73.7%)	49 (76.6%)	38 (70.4%)	
Alcohol	Never	15 (12.7%)	6 (11.1%)	9 (14.1%)	0.61
	Less than once a month	29 (24.6%)	13 (24.1%)	16 (25.0%)	
	2 or 4 times a month	63 (53.4%)	31 (24.1%)	32 (50.0%)	
	2 or 3 times a week	10 (8.5%)	3 (5.6%)	7 (10.9%)	
	4 or more times a week	1 (0.9%)	1 (1.9 %)	0	

**Table 2 vaccines-11-00618-t002:** Health Belief Model and HCW’s vaccination attitude by measles immunization status.

		Total HCW Respondents *n*(%)	Measles Immunized *n*(%)	Measles Not Immunized *n*(%)	*p* Value
		118	64 (54.2%)	54 (45.8%)	
Health Belief Model (Domains)					
Perceived susceptibility to measles infection					
	High level	86 (72.9)	47 (73.4)	39 (72.2)	0.882
	Medium-Low level	32 (27.1)	17 (26.6)	15 (27.8)	
Perceived severity of measles infection					
	High level	77 (65.3)	46 (71.9)	31 (57.4)	0.100
	Medium-Low level	41 (34.7)	18 (28.1)	23 (42.6)	
Perceived barriers to measles vaccination					
	High level	47 (39.8)	40 (62.5)	31 (57.4)	0.573
	Medium-Low level	71 (60.2)	24 (37.5)	23 (42.6)	
Perceived benefits of measles vaccination					
	High level	96 (81.4)	53 (82.8)	43 (79.6)	0.658
	Medium-Low level	22 (18.6)	11(17.2)	11 (20.4)	
Vaccination Beliefs					
Do you think that vaccinations for healthcare professionals are a prerequisite for their health?					
	Absolutely disagree	1 (0.9)	0	1 (1.9)	0.813
	Mostly disagree	2 (1.7)	1 (1.6)	1 (1.9)	
	Neither agree nor disagree	6 (5.1)	4 (6.3)	2 (3.7)	
	Fairly agree	35 (29.7)	19 (29.7)	16 (29.6)	
	Absolutely agree	74 (62.4)	40 (62.5)	34 (63.0)	
Do you think that vaccinations of health workers are an important tool to avoid absences from work?					
	Absolutely disagree	2 (1.7)	0	2 (3.7)	0.562
	Mostly disagree	5 (4.2)	2 (3.1)	3 (5.6)	
	Neither agree nor disagree	9 (7.6)	5 (7.8)	4 (7.4)	
	Fairly agree	46 (39.0)	25 (39.1)	21 (38.9)	
	Absolutely agree	56 (47.5)	32 (50.0)	24 (44.4)	
During your clinical practice, do you recommend vaccinations to your patients?					
	Yes	91 (77.1)	52 (81.3)	39 (72.2)	0.070
	Not	2 (1.7)	2 (3.1)	0	
	Sometimes	4 (3.4)	3 (4.7)	1 (1.9)	
	It is not my competence	17 (14.4)	4 (6.3)	13 (24.1)	
	I do not know	3 (2.5)	2 (3.1)	1 (1.9)	
	Other	1 (0.9)	1 (1.6)	0	
Immunization status against some exanthematous diseases					
Immunized against chickenpox (complete vaccination with two doses or acquired infection)					
	Yes	92 (77.9)	57 (89.1)	35 (64.8)	
	Not	26 (22.1)	7 (10.9)	19 (35.2)	
Immunized against mumps (complete vaccination with two doses or acquired infection)					
	Yes	71 (60.2)	56 (87.5)	15 (27.8)	<0.001
	Not	47 (39.8)	8 (12.5)	39 (72.2)	
Immunized against rubella (complete vaccination with two doses or acquired infection)					
	Yes	68 (57.6)	55 (85.9)	13 (24.1)	<0.001
	Not	50 (42.4)	9 (14.1)	41 (75.9)	

**Table 3 vaccines-11-00618-t003:** Univariable and multivariable analysis of factors associated with lack of measles immunization status.

		Crude OR	*p*	Adjusted OR	*p*
**Gender**	Male	ref		ref	
	Female	2.05	0.060	3.70	0.056
**Age (per year increment)**		0.82	0.311	1.18	0.648
**Smoke**	No	ref		ref	
	Yes	1.38	0.447	0.41	0.276
**Alcohol**	No	ref		ref	
	Yes	1.07	0.742	0.66	0.282
**Occupation Physician**	Yes	ref		ref	
	No	2.89	0.040	10.27	0.015
**Susceptibility to measles infection**	Low	ref		ref	
	High	0.94	0.882	0.89	0.893
**Severity of measles infection**	Low	ref		ref	
	High	0.53	0.100	1.13	0.872
**Barriers to measles vaccination**	Low	ref		ref	
	High	1.24	0.574	5.13	0.047
**Benefits of measles vaccination**	Low	ref		ref	
	High	0.81	0.659	1.04	0.967
**Immunized against chickenpox**	Yes	ref		ref	
	No	4.42	0.002	9.93	0.003
**Immunized against mumps**	Yes	ref		ref	
	No	18.20	<0.001	33.64	<0.001
**Immunized against rubella**	Yes	ref		ref	
	No	19.27	<0.001	10.12	0.002

## Data Availability

Data will be available after a motivated request to the corresponding author.

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
