# Peer review of "Measles Immunization Status of Health Care Workers: A Cross-Sectional Study Exploring Factors Associated with Lack of Immunization According to the Health Belief Model"

_vaccines, 2023, doi:10.3390/vaccines11030618_

Round 1

Reviewer 1 Report

I find interesting to explore the factors associated to the HCWs  low MMR vaccination rates, in order to point up barriers to vaccinaton that should be addressed.  Point up that most of  the barriers could be adressed by appropiate  protocols and guidelines to be followed by hospital vaccination units 

Introduction: 

How the hospital vaccinations unit offer MMR or other vaccines to HCWs?  The strategy should be explained in the introduction. HCWs could have refused vaccination or maybe they have not even  been offered vaccines. 

MM

MMR (measles, rubella and mumps) vaccine is administered in a unique vacination act during the first years of life all over the world; in this context I find inapropiate to ask separately if  someone receive measles, rubella or mumps vaccine. 

In order to obtain a more accurate information about the susceptibility status againts measles, some checking items, as documentation of vaccination, information from  vaccination registries should be included in the study. Using a questionnaire as the unique source of information,   could invalidate the results of the study  because of the recall bias. 

I dont think some habits, like taking mediaction regularly, smoke or the alcohol consumption are relate to the propensity to received measles vaccine. I suggest to remove this items from the analiys and the results 

Author Response

Reviewer 1

I find interesting to explore the factors associated to the HCWs  low MMR vaccination rates, in order to point up barriers to vaccinaton that should be addressed.  Point up that most of  the barriers could be adressed by appropiate  protocols and guidelines to be followed by hospital vaccination units 

Introduction: 

How the hospital vaccinations unit offer MMR or other vaccines to HCWs?  The strategy should be explained in the introduction. HCWs could have refused vaccination or maybe they have not even  been offered vaccines. 

Thank you so much for the question which allowed us to better explain the background of the investigation. The Vaccination Unit of the Palermo University Hospital is mainly dedicated to vaccination of HCWs and hospitalized patients. The vaccination unit works five days a week. Although MMR vaccination for HCWs is strongly recommended by “National Vaccine Prevention Plan”, “Italian law 119/2017” and “Ministerial Circular 25233/2017”, MMR vaccine remains non-mandatory (lines 55-58 and lines 177-185). Healthcare professionals should accept the measles vaccination recommendation as an ethical duty, to protect themselves and their patients.

In any case, at the end of the questionnaire administered to the HCWs there was a link to online booking to Vaccination Unit. These data were not reported in the study as very few HCWs booked an appointment to catch up on MMR vaccination.

MMR

MMR (measles, rubella and mumps) vaccine is administered in a unique vacination act during the first years of life all over the world; in this context I find inapropiate to ask separately if  someone receive measles, rubella or mumps vaccine. 

It is undoubtedly true that the MMR vaccine is administered in a single vaccination but, in our survey we investigated the immunization status against measles and other infection  diseases, regardless the vaccination status (i.e. both immunization related to primary infection and vaccination, please see lines 97 – 100) (Table 2). In particular, we did not ask separately about single vaccine  for measles, rubella or mumps, but we asked about the immunological status, considering that it is possible to get measles infection without getting mumps or rubella. We have taken your suggestion into account and  we specified better the association between “not being immunized for measles” and “not being immunized for other exanthematous diseases”, considering the lack of vaccination for MMR but also the lack of infections (lines 220-239).

In order to obtain a more accurate information about the susceptibility status againts measles, some checking items, as documentation of vaccination, information from  vaccination registries should be included in the study. Using a questionnaire as the unique source of information,   could invalidate the results of the study  because of the recall bias. 

It would be certainly appropriate to investigate the immunization status by consulting vaccination documentation and vaccination registries. Unfortunately, Sicily does not have a unique online vaccination register for citizens born after 2010 and this is certainly a major limitation for Sicilian public health routine vaccination practice. For people born before 2010 vaccinations were registered in a paper-based registry or in the vaccination card of each citizen. Furthermore, it was specified among the limitations of the study that the immunological status of HCWs was not verified with serological tests (lines 238 -239). However, according to the CDC recommendations for MMR vaccination in HCWs, serological screening for measles, rubella, or mumps immunity is not necessary in doubtful or undocumented cases of infection but two doses of vaccine should be administered (1).

  1. Centers of Disease Control and Prevention, https://www.cdc.gov/mmwr/preview/mmwrhtml/rr6204a1.htm

In addition to collecting information through the questionnaire, we also consulted the Vaccination Register of the Palermo Hospital Vaccination Unit that was based on vaccination reported in the vaccination card. We specified better the details on the eligibility criteria of the study population (lines 79 – 85).

I dont think some habits, like taking medication regularly, smoke or the alcohol consumption are relate to the propensity to received measles vaccine. I suggest to remove this items from the analysis and the results 

According to the literature, it is very frequent to investigate habits such as "smoking" or "drinking alcohol" because a more or less healthy lifestyle can be associated in a variable way with the adoption of a preventive practice, such as vaccination. For example, smokers frequently adopt a less healthy lifestyle and are also less likely to adhere to vaccinations. Obviously, it is not a specific association for measles vaccination, but it refers to the attitude towards prevention in general. Other investigations on factors related to vaccine acceptance have reported significant results for the association with "smoking" and other lifestyle habits (2) (3).

  1. Wu J, Li Q, Silver Tarimo C, Wang M, Gu J, Wei W, Ma M, Zhao L, Mu Z, Miao Y. COVID-19 Vaccine Hesitancy Among Chinese Population: A Large-Scale National Study. Front Immunol. 2021 Nov 29;12:781161. doi: 10.3389/fimmu.2021.781161.
  2. Remschmidt, C., Fesenfeld, M., Kaufmann, M. A., Delere, Y. Sexual behavior and factors associated with young age at first intercourse and HPV vaccine uptake among young women in Germany: implications for HPV vaccination policies; doi: 10.1186/1471-2458-14-1248.

Reviewer 2 Report

This is a timely manuscript in light of rising vaccine hesitancy globally, including in Europe and the important role healthcare providers play in educating patients and limiting nosocomial outbreaks.  However, I'm concerned that the convenience sampling approach is insufficient to understand under-immunization among University Hospital of Palermo by profession.  Also, you include prior immunization against mumps, rubella, and varicella disaggregated by measles immunization status, though MMR has been a combined vaccine in Italy since the early 1990s, while varicella wasn't introduced nationally until 2015.  Participants >30 yrs may not have had the opportunity to receive two doses of mumps or varicella vaccine.  This needs to be accounted for in the analysis and results.  

Additional comments by section:

Background - critical to discuss the current Italian immunization schedule, i.e. MMRV/MMR+V at 13-15 months and 6 yrs of age.  Why is 2nd dose MCV coverage shared for those 18 yrs of age, for more contemporary estimates should closer to 6 yrs of age, when given.  Per WHO, MCV2 coverage was 86% in 2021 - https://immunizationdata.who.int/pages/coverage/mcv.html?CODE=ITA&ANTIGEN=MCV2&YEAR=

Materials and Methods - Critical to mention the number of HCWs at the hospital by profession, the # with a telephone number, and then the 122 with consent attempted and 120 enrolled, to understand your overall response rate, and the rates by profession.   Also, if both text messages and e-mails were sent, why did a respondent require a phone number?  Couldn't e-mail or phone number suffice.  In methods, it should be mentioned what p value designated significance, e.g. <0.05?     

Results - With 63% of your sample being medical residents or medical students, I'd suggest reporting median age as well as the mean age of 31 is likely skewed by a few older staff.  Having recruited only 7 nurses, it limits an understanding of non-physician attitudes toward vaccination.   Since p=0.056, the finding that females are less likely to be immunized against measles than males in the multivariate analysis may not be significant unless the p value cutoff is mentioned in methods.  In the health beliefs model table, disagree is misspelled twice.   Again, analyzing immunization against varicella, mumps, and rubella independently would be difficult to determine if received combined vaccine.       

Discussion:  It should be mentioned that verification of reported vaccination status through medical records or immunization cards was not completed, another limitation. 

Author Response

Reviewer 2

This is a timely manuscript in light of rising vaccine hesitancy globally, including in Europe and the important role healthcare providers play in educating patients and limiting nosocomial outbreaks.  However, I'm concerned that the convenience sampling approach is insufficient to understand under-immunization among University Hospital of Palermo by profession.  Also, you include prior immunization against mumps, rubella, and varicella disaggregated by measles immunization status, though MMR has been a combined vaccine in Italy since the early 1990s, while varicella wasn't introduced nationally until 2015.  Participants >30 yrs may not have had the opportunity to receive two doses of mumps or varicella vaccine.  This needs to be accounted for in the analysis and results.  

Additional comments by section:

Background - critical to discuss the current Italian immunization schedule, i.e. MMRV/MMR+V at 13-15 months and 6 yrs of age.  Why is 2nd dose MCV coverage shared for those 18 yrs of age, for more contemporary estimates should closer to 6 yrs of age, when given.  Per WHO, MCV2 coverage was 86% in 2021 - https://immunizationdata.who.int/pages/coverage/mcv.html?CODE=ITA&ANTIGEN=MCV2&YEAR=

Thank you for your suggestion. It is certainly appropriate to mention the current measles vaccination schedule and we have added this information at lines 44-48. Furthermore, 18-year-olds represent a cohort of population which is the closest in age to our sample (i.e. students of the health professions and professional health workers). For this reason, we thought it was appropriate to report the vaccination coverage of an adult population, such as that represented by 18-year-olds.

Materials and Methods - Critical to mention the number of HCWs at the hospital by profession, the # with a telephone number, and then the 122 with consent attempted and 120 enrolled, to understand your overall response rate, and the rates by profession.   Also, if both text messages and e-mails were sent, why did a respondent require a phone number?  Couldn't e-mail or phone number suffice.  In methods, it should be mentioned what p value designated significance, e.g. <0.05?     

Thank you for your observation which  allowed us to better explain the methods of investigation. About 2,000 HCWs work at the University Hospital of Palermo; we invited 900 HCWs to respond to the questionnaire because they were the only people with an available telephone number and email address. Both text messages and emails were sent inviting HCWs to answer to the questionnaire. The text of the message and emails was the same and included  the link to the questionnaire created online. A limit of only one answer for each participant had been foreseen and imposed. We chose to send the message twice, first via phone number and, four months later, via email as a reminder, to have the possibility of enrolling HCWs through two different means of communication.

Regarding the significance value, we added the following sentences in the method section: “A multivariable logistic regression model was used to analyse the covariates associated at univariable analysis with a p value lower than 0.10 and for a priori confounding variables.” (lines 111-114).

Results - With 63% of your sample being medical residents or medical students, I'd suggest reporting median age as well as the mean age of 31 is likely skewed by a few older staff.  Having recruited only 7 nurses, it limits an understanding of non-physician attitudes toward vaccination.   Since p=0.056, the finding that females are less likely to be immunized against measles than males in the multivariate analysis may not be significant unless the p value cutoff is mentioned in methods.  In the health beliefs model table, disagree is misspelled twice.   Again, analyzing immunization against varicella, mumps, and rubella independently would be difficult to determine if received combined vaccine.   

Thanks for your suggestions. The mean age of the sample is mentioned in line 120 and, immediately afterwards, in line 122 the composition of the sample is mentioned, with 33% medical residents and 30% students of health professions.  The mean age has been reported since the age of the HCWs interviewed is a normally distributed variable.

Work occupation was evaluated, in the logistic analysis, by distinguishing "non-Physician" “HCWs” and "Physician". Therefore, the group of "non-Physicians" is numerically well represented as they include nurses, social-health workers, students…

With regards to the p value = 0.056, in relation to the difference between the two sexes, we considered it statistically significant as we have conventionally accepted a significance for p values ≤ 0.10, as reported in the lines 111 – 114.

We checked and corrected the misspelling in table 2.

In our survey, we investigated the immunization status against measles and other infection  diseases, regardless the vaccination status (i.e. both immunization related to primary infection and vaccination, please see lines 97 – 100) (Table 2). In particular, we did not ask separately about single vaccine  for measles, rubella or mumps, but we asked about the immunological status, considering that it is possible to get measles infection without getting mumps or rubella. Thank you for your suggestion, because it allowed us to better specify the association between “not being immunized for measles” and “not being immunized for other exanthematous diseases” , considering the lack of vaccination for MMR but also the lack of infection.

Discussion:  It should be mentioned that verification of reported vaccination status through medical records or immunization cards was not completed, another limitation. 

It would be certainly appropriate to investigate the immunization status by consulting vaccination documentation and vaccination registries. Unfortunately, Sicily does not have a unique online vaccination register for citizens born after 2010 and this is certainly a major limitation for Sicilian public health routine vaccination practice. For people born before 2010, vaccinations were registered in a paper-based registry or on the vaccination card of each citizen. However, we have taken  your kind suggestion into consideration and we added this critical element at the end of “Discussion” section (lines 247 - 250). Moreover, according to the CDC recommendations for MMR vaccination in HCWs, serological screening for measles, rubella, or mumps immunity is not necessary in doubtful or undocumented cases of infection but two doses of vaccine should be administered (1).

Reviewer 3 Report

This is a very good article dealing most with medical students and physicians and measles vaccination interests. The goal is noble, to protect HCW against measles or promote mandatory vaccination. The manuscript is well written and the analysis well defined and presented. The discussion is a very adequate and extensive.  I noted some problems in the sample. 

The sample is the problem and the main problem of the article. There are only 12/118 people not "physician, resident or medical student" thus I believe that Health care workers denomination cannot be applied to their sample.  The best name would be medical personal and students, and check the analysis without those non-medical HCW.  Comparing non-medical with medical beliefs are important.  How would be the analysis without those non-medical members?  Young medical staff  unprotected against measles, one of most transmissible viral disease, probably would be a problem for medical schools.  

I could suggest several other sorting systems for their analysis of the data, but the authors must search for their adequate way for looking to this problem. 

Author Response

This is a very good article dealing most with medical students and physicians and measles vaccination interests. The goal is noble, to protect HCW against measles or promote mandatory vaccination. The manuscript is well written and the analysis well defined and presented. The discussion is a very adequate and extensive.  I noted some problems in the sample. 

The sample is the problem and the main problem of the article. There are only 12/118 people not "physician, resident or medical student" thus I believe that Health care workers denomination cannot be applied to their sample.  The best name would be medical personal and students, and check the analysis without those non-medical HCW.  Comparing non-medical with medical beliefs are important.  How would be the analysis without those non-medical members?  Young medical staff  unprotected against measles, one of most transmissible viral disease, probably would be a problem for medical schools.  

I could suggest several other sorting systems for their analysis of the data, but the authors must search for their adequate way for looking to this problem. 

Thank you for appreciating the paper and for highlighting the importance of the topic. We are aware of the sample related limitations as reported in the manuscript (lines 263 – 266). However, the study on this small convenience sample showed significant results that could be useful for improving measles immunization in HCWs. Thank you for the suggestion. We have changed the classifications of HCWs according to their occupation: physician (n=23), medical resident (n=39), medical student (n=36), other healthcare workers (n=20) (table 1, lines 132-135). We believe it is important to not remove other healthcare professionals because overall they contribute significantly to the analyses.

Round 2

Reviewer 1 Report

dear authors, 

Your responses do not cover the main obstacles that I suggest you after my first revision. 

Author Response

dear authors, 

Your responses do not cover the main obstacles that I suggest you after my first revision. 

We are very sorry that we have not clarified your doubts. We would be glad if you can suggest us what was missing in our answers. However, we have made changes to our paper by adding the response rate of the interviewed HCWs (lines 121-122), the history of the MMR vaccine offer in Italy (lines 44-50) ( lines 61-62), specifying the cut off of significant multivariate results (lines 152-156) and modifying the discussions (243-244). We hope the changes can increase the quality of the paper.

Reviewer 2 Report

Thank you for accepting many of my suggested revisions, though others remain unresolved.  Here's a list:

Background

1.  Since you include questions about varicella, mumps, and rubella immunization in the background you should mention the current and historical vaccine requirements, i.e. MMR has been a combined vaccine in Italy since the early 1990s, while varicella wasn't introduced nationally until 2015.     

Methods

2.  Since only 900 of 2,000 HCWs have a phone/e-mail on file, you should say something about how these populations differ, i.e. were those with contact information older or more likely doctors or students/residents?

3. Since varicella wasn't introduced nationally until 2015, most respondents would not have received, so would be interested in restricted analysis, removing those who would not have had the opportunity to receive varicella.     

4.  Fine to use a cut-off of p=0.10 for inclusion in the multivariable, but significance in the multivariate model should be P < 0.05,  

Results 

3. The response rate (120/900 = 13%) should be included in the first sentence of results.   

4.  You mention that "In our survey, we investigated the immunization status against measles and other infection  diseases, regardless the vaccination status (i.e. both immunization related to primary infection and vaccination, please see lines 97 – 100)  but unsure how you would investigate immunization status regardless of vaccination status, as they're synonymous.    

Discussion

You mention that "The lack of immunization against these diseases was due to the lack of vaccination and the lack of immunity acquired following a natural infection." though you only asked about "perceived susceptibility to measles infection" as opposed to prior natural infection history. 

You need a reference for this statement "the non-occurrence of measles infection and other infectious diseases could be related precisely to the adoption of these hygienic-sanitary standards" in lines 232-3. 

Author Response

Thank you for accepting many of my suggested revisions, though others remain unresolved.  Here's a list:

Background

  1. Since you include questions about varicella, mumps, and rubella immunization in the background you should mention the current and historical vaccine requirements, i.e. MMR has been a combined vaccine in Italy since the early 1990s, while varicella wasn't introduced nationally until 2015.  

Thank you for your suggestion.  It is important to mention the history of the introduction of the MMR vaccine in Italy and the current vaccination offer to the population (lines 44-50) (lines 61-62) .

Methods

  1. Since only 900 of 2,000 HCWs have a phone/e-mail on file, you should say something about how these populations differ, i.e. were those with contact information older or more likely doctors or students/residents?

We try to better explain the process adopted for including respondents in the study. First of all, we had only the phone/e-mail of the HCWs without any other demographic information. Subsequently, we send the SMS and e-mail to have consent to be contacted. Of the 2,000 HCWs 1,098 didn’t reply to the e-mail/phone messages and 2 did not give the consent to collect data. So we do not have information on the difference between enrolled and not enrolled people. We understand that should be relevant for the difference in term of age, occupation but also other factors should make difference in immunization against MMR immunization as standard procedures adherence or modifiable habits. We inserted this further limitation in the discussion section: “Furthermore, the enrolment procedure based on SMS and e-mail that could have included HCWs more suitable to answer on line questionnaire and the small sample size is not representative of all Italian HCWs. This could limit the generalizability of data.”

  1. Since varicella wasn't introduced nationally until 2015, most respondents would not have received, so would be interested in restricted analysis, removing those who would not have had the opportunity to receive varicella.     

Varicella vaccination was introduced in Sicily about 10 years earlier than in the rest of Italy. In January 2003, a universal vaccination program against varicella was launched in Sicily, which was included in the regional vaccination calendar together with vaccinations against diphtheria, tetanus, poliomyelitis, hepatitis B, pertussis, Haemophilus influenzae type b, measles, mumps, and rubella, offered free to all children (1). Even if the healthcare workers interviewed did not receive varicella vaccine as infants (13 - 15 months) they could have catch up the vaccination as adults, as it is strongly recommended for all occupationally exposed workers (lines 57-58) (lines 178-180). Furthermore, we investigated the immunization status against varicella, including both vaccination and infection occurred (Table 2). The respondents could have acquired natural immunity from infection. I hope I have answered your question comprehensively.

  1. Giammanco, G., Ciriminna, S., Barberi, I., Titone, L., Lo Giudice, M., Biasio, L.R. Universal varicella vaccination in the Sicilian paediatric population: rapid uptake of the vaccination programme and morbidity trends over five years. Euro Surveill. 2009 Sep 3;14(35):19321.

  1. Fine to use a cut-off of p=0.10 for inclusion in the multivariable, but significance in the multivariate model should be P < 0.05, 

Thank you for your suggestion. “A multivariable logistic regression model was used to analyze the covariates associated at univariable analysis with a p value lower than 0.10 and for a priori confounding variables” (lines 114-116). A cut off p < 0.10 was used for inclusion in the multivariate analysis; but  the statistically significant results of the multivariable model were p < 0.05. After your suggestion, we thought it appropriate to clarify that the result referring to the lowest immunization rate in female sex was bordering on significance (lines 152 – 156) (lines 243 – 244).

Results 

  1. The response rate (120/900 = 13%) should be included in the first sentence of results.   

Thanks for your suggestion. We have added the response rate, as you recommended (lines 121 – 122).

  1. You mention that"In our survey, we investigated the immunization status against measles and other infection  diseases, regardless the vaccination status (i.e. both immunization related to primary infection and vaccination, please see lines 97 – 100)  but unsure how you would investigate immunization status regardless of vaccination status, as they're synonymous.    

We investigated the immunization status of healthcare workers by assessing natural infection occurred and the vaccination status. A person who contracted the natural infection is immunized against a new measles infection, even though he has not received the vaccination. Therefore, it is possible to be immunized without being vaccinated.

Discussion

You mention that "The lack of immunization against these diseases was due to the lack of vaccination and the lack of immunity acquired following a natural infection." though you only asked about "perceived susceptibility to measles infection" as opposed to prior natural infection history. 

Using the constructs of the Health Belief Model we investigated the perceived susceptibility to measles infection. In detail, the questions concerned the increased risk to contract measles associated with occupational exposure. Therefore, we investigated a susceptibility related to the risk of infection and contagion in the hospital setting. The “susceptibility domain of HBM” is unrelated to primary infection that may have occurred in childhood, but it concerns the perception of being more exposed to the possibility of contagion.

You need a reference for this statement "the non-occurrence of measles infection and other infectious diseases could be related precisely to the adoption of these hygienic-sanitary standards" in lines 232-3. 

Thank you for the suggestion. The bibliographic reference may help strengthen the hypothesis (lines 237-238).